# Convergent approach for direct cross-coupling enabled by flash irreversible generation of cationic and anionic species

Hiroki Soutome [1,2], Hiroki Yamashita[1], Yutaka Shimizu[1], Masahiro Takumi[1], Yosuke Ashikari [1] & Aiichiro Nagaki [1] ✉

In biosynthesis multiple kinds of reactive intermediates are generated, transported, and reacted across different parts of organisms, enabling highly sophisticated synthetic reactions. Herein we report a convergent synthetic approach, which utilizes dual intermediates of cationic and carbanionic species in a single step, hinted at by the ideal reaction conditions. By reactions of unsaturated precursors, such as enamines, with a superacid in a flow microreactor, cationic species, such as iminium ions, are generated rapidly and irreversibly, and before decomposition, they are transported to react with rapidly and independently generated carbanions, enabling direct C-C bond formation. Taking advantage of the reactivity of these double reactive intermediates, the reaction take place within a few seconds, enabling synthetic reactions which are not applicable in conventional reactions.

Proteins are organic molecules with unique functions. In organisms, proteins such as enzymes are intertwined to perform organic reactions and synthesize biogenic molecules, which are essential for maintaining life[1]. Such biosynthesis, which takes full advantage of organic molecules, is one of the ideal goals in organic chemistry. During biosynthesis, multiple kinds of "reactive intermediates" are generated, transported, and reacted across different parts of living organisms[2]. The mobility of these intermediates enables high selectivity, even though the reactions occur at rates close to diffusion-limited reactions. In such reactions, the selectivity depends on the rate of transport of the reactants instead of their reaction kinetics. Thus, biological reactions are often considered to occur under ideal reaction conditions, and it is expected that excellent reaction selectivity using active intermediates can be achieved by approaching such conditions in organic synthetic reactions[3].

We studied the organic synthesis mediated by highly reactive intermediates. In particular, we utilized flow microreactors to develop methods for the fast generation and reactions of short-lived anionic species, which ordinal batch reactors cannot handle[4]. In these methods, unstable anionic species, particularly carbanions, are rapidly and irreversibly generated and immediately react with electrophiles. This leads to the complete conversion of starting materials into highly reactive intermediates, which enable subsequent fast reactions. "Flash" chemistry[5], which utilizes the anionic reactive intermediates in rapid reactions, has enabled various chemical transformations that traditional methods cannot achieve.

Cations, which are unstable species as well as anions, are also important reactive intermediates, with organic cations being the most significant for organic synthesis[6]. Among a wide range of conventional methods to generate organic cations, such as acid-promoted methods[7], diazotization[8], and chemical oxidation[9], those involving Lewis acids are the most frequently utilized. Since this method reversibly generates cations, the concentration of the cations is low, making the reaction slow. To overcome this drawback, organic electrosynthesis[10–12], which generates relatively stable organic cations irreversibly, has been developed. Owing to their high reactivity, electrochemically generated cations can react rapidly with nucleophiles. However, in most electrochemical reactions, anodic oxidation requires several hours to generate cationic species[13], resulting in a slower reaction. We envisaged that if highly unstable and reactive cationic species were generated rapidly, this would lead to a new synthetic approach, where multiple reactive intermediates are utilized in

[1]Department of Chemistry, Graduate School of Science, Hokkaido University, Sapporo, Hokkaido, Japan. [2]Yokohama Technical Center, AGC Inc, Yokohama, Kanagawa, Japan. ✉e-mail: nagaki@sci.hokudai.ac.jp

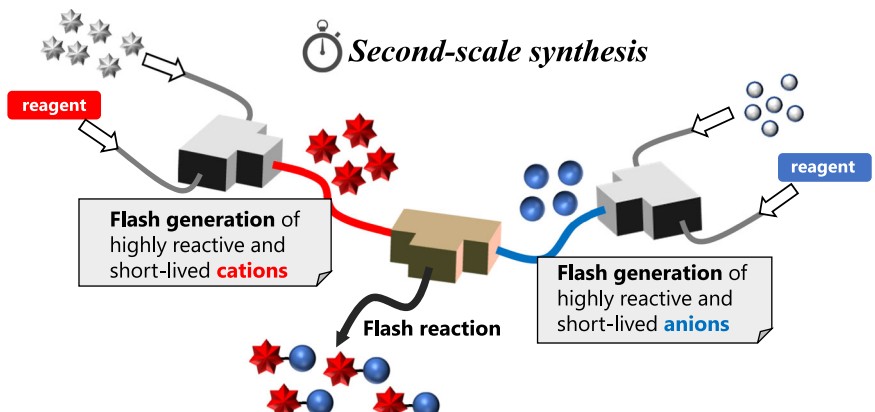

**Fig. 1 | Second-scale synthesis via flash generation of reactive intermediates.** A schematic for convergent approach for direct cross-coupling. In the left reactor, short-lived cations are generated, and in the right reactor, short-lived anions are generated. Those cationic and anionic species proceed to flash reactions before they decompose.

different locations, mimicking an idealized reaction system, such as biosynthesis.

This article reports a proof-of-concept study that uses flowmicro methods to demonstrate cationic species' irreversible and rapid generation, followed by their reactions with rapidly generated carbanions and a direct cross-coupling reaction to generate carbon-carbon bonds within seconds. These achievements showcase a simultaneous use of multiple highly-reactive intermediates, which transform the "flash chemistry" into "flash synthesis," leading to a new synthetic chemistry (Fig. 1).

## Results

### Cationic species generation in batch and flow reactor

To achieve the rapid and irreversible generation of cationic species, we focused on the reactions of vinyl compounds with a strong Brønsted acid as a strong proton donor (Fig. 2)[14]. Among a variety of methods for generating cationic species, the use of strong acid may provide a rapid generation of those unstable species without the specific equipment such as flow electrolysis devices. We selected enamines as precursors for iminium ions, and trifluoromethanesulfonic acid (TfOH, pKa 2.6 in acetonitrile[15] as the proton source because of its high acidity and ease of handling[16]. The reaction of enamines with strong acids generates iminium ions; however, since the enamines may work as nucleophiles, the generated cations easily react with the enamines, resulting in dimerization. Actually, the reaction of enamine **1a** with TfOH in a batch reactor followed by the addition of allyltrimethylsilane afforded a small amount of the desired product **3**, and dimer **4** was obtained as the major product at each reaction temperature (Fig. 2, Method A and B, for yields of dimer **4** see Supplementary Table 2 for detail). The reactions of **1a** with TfOH in the presence of allyltrimethylsilane showed similar results (Fig. 2 and Method C and D). However, altering the sequence of reagent addition affected the result: methods A and C, where the solution of **1a** was added to the solution of TfOH, resulting in a higher yield of **3** than the reverse-drop method (Method B and D) respectively. This tendency suggests the mixing efficiency affects the reaction efficiency. The batch reactions with 1.0 equivalent of TfOH also resulted in low yields (see Supplementary Information). The above investigations, where dimer **4** was obtained more abundantly than the desired product **3**, indicate that the generated iminium ion **2a** rapidly reacted with the surrounding enamine **1a**. Thus, to achieve the selective generation of **2a** for its reactions with nucleophiles, **1a** must be converted to **2a** before the dimerization.

To improve reaction selectivity, flow microreactors have attracted considerable attention for decades[17,18]. In particular, their fast mixing characteristics remarkably affect fast reactions[19,20]. We have demonstrated that the fast mixing of flow micromixers enabled selective reactions of a sulfur cation with styrenes before their cationic polymerization[21]. Based on this idea, we envisaged that flow microreactor would achieve the selective generation of iminium ion **2a** before their reaction with **1a**, and thus, investigated the reactions of enamines, TfOH, and allylsilanes in a flow microreactor (Fig. 2a). Surprisingly, the desired product was obtained in high yield, especially at a higher flow rate (Fig. 2b, bars) This indicates that the condition improving the mixing efficiency is beneficial for this transformation[22], which was supported by the reactions using other micromixers. The reactions with a thinner and sharper-angled micromixer, which has a better-mixing efficiency[23], showed higher yield.

For a deeper understanding of the mixing efficiency, we explored the Villermaux–Dushman method, which assesses the mixing efficiency of micromixers and estimates the mixing time of aqueous medium[24]. Based on Yin's condition[25], we estimated the mixing time of the solutions introduced into the V-shaped 250 μm micromixer (Fig. 2b, dots). The results clearly showed the difference; the condition with the smallest flow rate (2.5 mL/min) had a longer mixing time (more than 1 millisecond), whereas that of other conditions was less than 0.1 millisecond. Although these values are those of aqueous medium, the tendency and scale of the mixing time must be the same. The T-shaped mixers also showed longer mixing times than those of the V-shaped mixer. Although an advanced analysis such as CFD simulation must be necessary to gain an accurate one, we calculated the Damköhler number of these conditions to be below 0.1 (see Supplementary Information). This indicates that the mass transport is enough faster than the reaction, which is supported by the fact that a higher flow rate than 10 mL/min did not change the product yields[26]. The generation of dimer **4** supports this idea. The best results under the batch condition (Method C, −40 °C) showed non-negligible amount of **4**[27], which is derived from the reaction of **1a** with **2a**, whereas the flow condition (V250, 20 mL/min) showed less amount of **4** (Fig. 2c). Thus, these investigations proved that the reactions with short mixing time can trigger the desired reaction between TfOH and enamine **1a** before it reacts with **2a**. These results indicate that the flow microreactor plays a crucial role in the rapid generation of the iminium ion by adding superacids.

Accurate controllability of the reaction time[28,29] and temperature is also a significant characteristic of flow microreactors[30,31] We have previously reported that the controllability of flow microreactors can control various unstable and reactive species[32]. Inspired by this previous success, we screened the reaction time in the range of decamilliseconds to seconds and the reaction temperature. The results are summarized as the contour map in the left part of Fig. 2d, revealing

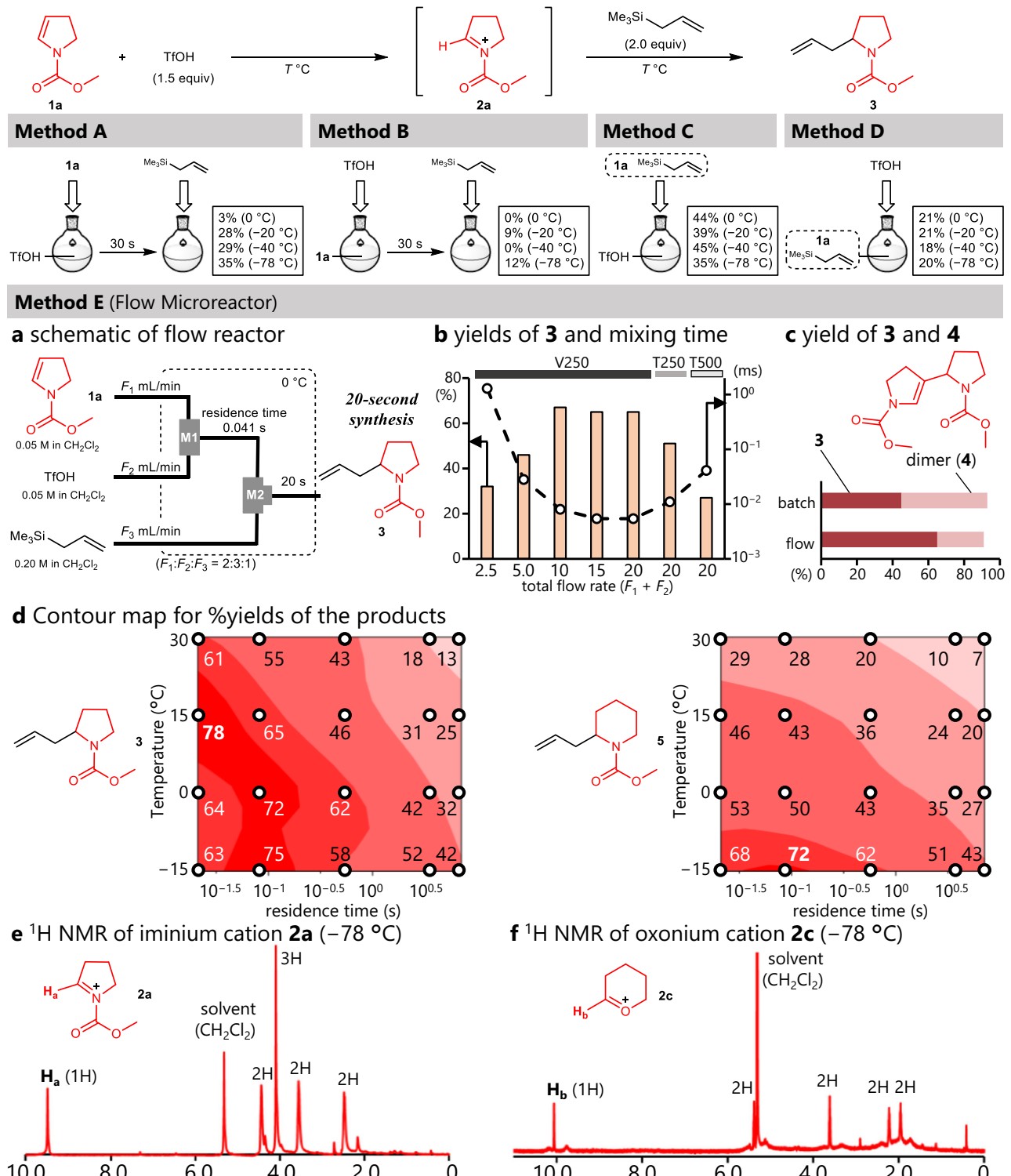

**Fig. 2 | Generation and reaction of onium cations in a batch and flow system.** [a]Methods (**A**–**D**) were done in a batch system, and Method E was done in a flow system composed of two micromixers. Yields were determined by GC. **a** Schematic for a flow reactor. **M2** is T-shaped mixer with 250 μm inner diameter. **b** Dependency of yields of **3** and mixing time on flow rates and mixer **M1**. The reaction time between **1a** with TfOH is 0.041 s. For mixers, the alphabet (V and T) denotes the shape of the mixer, whereas the number denotes its inner diameter (μm). bars: yield (%), dots: mixing time (millisecond) determined by Villermaux–Dushman method.

**c** Yields of **3** and **4** (dimer) determined by GC. Batch: method C at −40 °C, flow: V250, 20 mL/min. **d** Contour maps of yields of **3** (left) and **5** (right) using a V-250 mixer. Horizontal axis: residence time (s), vertical axis: temperature (°C), the designated number on the circle dot: yield of **3** and **5** (%). **e**, **f** Low-temperature NMR analyses of **2a** (**e**) and **2c** (**f**). The cationic species were generated in a flow microreactor using $CD_2Cl_2$ as a solvent and were flowed into an NMR sample tube cooled at −78 °C. [1]H NMR analyses were done at −78 °C.

that the reaction condition with 21 milliseconds of the reaction time and 15 °C of the reaction temperature affords the highest yield of this transformation. This map also indicates that conditions with a lower temperature and shorter reaction time gave a lower product yield, suggesting that these conditions are insufficient for the generation of the iminium ion. In addition, the lower yield at higher temperatures and longer reaction times could be derived from the decomposition of the iminium ion, such as the β-elimination of the generated iminium to afford the starting enamine. Although **1a** was not detected after the reactions, the amount of dimer **4**, which is generated from **1a** with iminium ion **2a** was increased when the reaction time and temperature increased. This indicates that at such conditions, **2a** was decomposed to **1a** (Supplementary Table 6 for details). Thus, this contour map can be regarded as a visualization of the stability and reactivity of the generated cation (**2a**). We then investigated the reaction mediated by a 6-membered ring iminium ion (**2b**) and determined the yield of the corresponding product **5** (Fig. 2d, right). Comparing these two contour maps shows that the pyrrolidine-type product **3** can be obtained in higher yields at higher temperatures and longer reaction times.

The irreversible generation of the cationic species was confirmed using a low-temperature NMR study. The reacting solution emitted from the flow reactor was captured in an NMR test tube cooled at −78 °C, and immediately its $^1$H NMR was measured at the same temperature. The NMR chart shows a highly deshielded proton $H_a$ (9.49 ppm, shown in Fig. 2e), indicating the generation of iminium ion **2a** as well as oxocarbenium ion **2c** (Fig. 2f, $H_b$ was appeared at 10.07 ppm). Notably, these NMR charts did not show the precursors (enamine **1a** for **2a** and acetal **1c** for **2c**), significantly supporting the irreversible generation of the iminium and oxonium cations. It is noteworthy that the irreversible generation of the iminium ion is crucial for its reaction. When using weaker acids such as benzoic acid (p$K$a 22), acetic acid (p$K$a 24), and trifluoroacetic acid (p$K$a 12 in acetonitrile)[33], their reactions with **1a** and allyltrimethylsilane in the flow microreactor did not afford **3**.

## Reactions of cations with nucleophiles and direct cross−coupling reaction

After establishing the flash generation of the cationic species, we investigated their reactions with various carbon nucleophiles (Table 1). C-C bond formed with neutral nucleophiles resulted in good to high yields of the corresponding products (entries 1–9). These nucleophiles are incompatible with conventional methods for reversible cation generation. In addition, the cations were generated within 82 ms, much faster than conventional electrochemical methods[34], resulting in a reaction time of only 20 s.

We attempted to develop reactions between multiple reactive species using the flash generation of cationic species and their reactions with neutral nucleophiles. Our flow strategy aimed to enable a reaction between the cationic species and carbanions, providing a direct cross-coupling involving C-C bond formation. As a feasibility study, we investigated the reactions of the iminium cation with alkyl metal species as carbanions ($n$Bu$^-$) and different counter ions (Li$^+$, MgCl$^+$, and ZnCl$^+$; Table 1, entries 10–12). Direct cation-anion coupling afforded the desired product **13**, where a sp$^3$-sp$^3$ C-C bond was formed in high yield by virtue of the fast mixing and precise time control together with the precise temperature control of the flow systems. Surprisingly, these cross-coupling reactions were completed within 0.069 s owing to the high reactivities of both the cationic species and carbanions. The flow microreactors enabled the direct cross-coupling of the reactive species that selectively proceeded without a side reaction, such as the β-elimination of the cation and protonation of the anions by the solvent (dichloromethane). On the other hand, we tested the reaction of anodically generated **1a** with $n$-butyllithium in a batch reactor, and found no desired product **13**, presumably because that the abundance of the supporting electrolyte in the surroundings led to

the side reaction with $n$BuLi. As the electronegativity of the counter ion increased (Li:0.98, Mg:1.31, and Zn:1.65), the flow reaction yield decreased, suggesting that reactive anions are suitable for direct reactions with the cationic species. Moreover, the reaction of the lithium species required the shortest time to complete (0.36 s). This suggests that because of its high reactivity, $n$-BuLi could react with the onium cation before decomposition. Additionally, the most effective way to achieve rapid reactions is through the direct reaction of cationic species with carbanions bearing a lithium counter ion.

## Convergent approach of double short-lived intermediates

Finally, we attempted to establish a biosynthesis-inspired reaction system in which multiple reactive intermediates were generated and reacted in flow condition. We have reported a linear reaction integration in which several intermediates are generated and reacted individually[35]. Based on the concept of reaction integration[36], we designed a convergent-type integration that generated cationic species irreversibly at one location and carbanions at another location (Table 2), enabling the simultaneous utilization of multiple reactive intermediates. As carbanions, aryl lithiums, which were generated from the corresponding aryl bromides with $n$BuLi in the flow microreactor[29], were utilized for this transformation. The coupling of the iminium and oxonium cations with the aryl anions in a flow microreactor resulted in the formation of sp$^3$-sp$^2$ carbon-carbon bonds and the synthesis of the coupling products in good to high yields (entries 1–12). Various aryl anions, including ones cannot be utilized in batch reactors due to their short lifetime (entry 3 and 10)[37], were used in the reaction with pyrrolidine-type iminium cations **2a**, **2d**, and **2e** (entries 1–10). Moreover, this convergent reaction demonstrated some examples that were not achieved by conventional methods: the mono-selective coupling of dibromoarenes (entry 4)[38], in which transition-metal-catalyzed cross-coupling is seldom achieved, and the reaction at the meta-position of an electron-donating group (entry 6), whose selectivity is prohibited in Friedel-Crafts-type reactions[39]. Iminium cations bearing other protecting groups, *tert*-butoxycarbonyl (Boc) and allyloxycarbonyl (Alloc), which can be easily deprotected under acidic conditions, were used in this reaction system (entries 8–10). Especially, by virtue of the high reactivity, the coupling reaction within one second was demonstrated (entry 10). The reactions of more unstable cations also afforded the products good to high yields (entries 11 and 12). These results demonstrate the effectiveness of this rapid generation and reaction concept, resulting in direct cross-coupling of various substrates.

## Direct cross-coupling with sp carbon

In addition to the flash formation of sp$^3$-sp$^3$ and sp$^3$-sp$^2$ C-C bonds, we investigated the cross-coupling involving sp$^3$-sp C-C bond formation[40]. The sp anions were generated from the terminal alkynes and $n$-butyllithium in the flow reactor, which subsequently reacted with the iminium cations to afford the coupling products in good to high yields (Table 2, entries 13–21). Iminium cations with increased instability (entries 20 and 21) and sp anions bearing electrophilic functionalities (entries 14–16)[41] were also applicable in this convergently integrated reaction. Because traditional reactions are inefficient for such transformations, introducing alkynyl groups to the α-position of pyrrolidines bearing electron-withdrawing groups in this rapid convergent reaction demonstrated great synthetic utility. As well as the iminium ions, this integrated flow system allowed the generation of thionium ion **2 f**, which reacted with sp anion to afford the coupling product in a good yield (entry 22).

Further investigation of sp$^3$-sp coupling showed that repeated lithiation of the alkyne and sp$^3$-sp coupling yielded unsymmetrical alkynes (Fig. 3). The first alkynylation reaction proceeded rapidly, and after deprotection and isolation, the next alkynylation completed within 2.8 s after the activation of the precursor. This series of

**Table 1 | Reactions of cationic species with nucleophiles in a flow microreactor[a]**

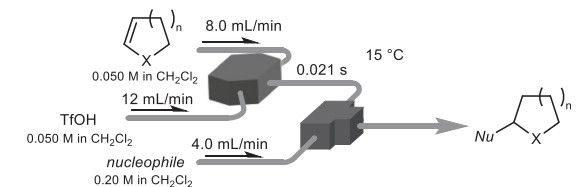

| entry | cation precursor | cation | nucleophile | product | total reaction time (s) | yield (%) |
|---|---|---|---|---|---|---|
| 1 | 1a | 2a | Me₃Si~~allyl~~ | 3 | 20 | 78 |
| 2 | | | OSiMe₃ cyclohexene | 6 | 20 | 60 |
| 3 | | | MeO–OSiMe₃ | 7 | 20 | 70 |
| 4 | | | OSiMe₃ | 8 | 20 | 83 |
| 5[b] | | | Ph–OSiMe₃ | 9 | 20 | 82 |
| 6[c] | 1b | 2b | Me₃Si~~allyl~~ | 5 | 21 | 72 |
| 7[c] | | | Ph–OSiMe₃ | 10 | 21 | 81 |
| 8[c] | | | OSiMe₃ | 11 | 21 | 82[d] |
| 9[e] | 1c | 2c | Ph–OSiMe₃ | 12 | 21 | 54 |
| 10[f] | 1a | 2a | ZnCl | 13 | 1.3 | 33 |
| 11[f] | | | MgCl | 13 | 1.3 | 65 |
| 12[f] | | | Li | 13 | 0.36 | 72 |

[a]Flow microreactor composed of a V-shaped micromixer (inner diameter: 250 μm) and a T-shaped one (inner diameter: 250 μm) was used. Yields were determined by GC.
[b]Temperature: 0 °C.
[c]Temperature: −15 °C, residence time: 0.082 s.
[d]Isolated yield.
[e]Temeprature: −25 °C, residence time: 0.082 s. 4 eq. of TfOH was used. See SI for details.
[f]Nucleophiles are dissolved in n-hexane or THF. Flow rate of **1a**: 6.7 mL/min, that of TfOH: 13.3 mL/min, that of nucleophiles: 3.4 mL/min, temperature: −10 °C, retention time: 0.29 s. See SI for details.

## Table 2 | Direct cross-coupling of sp³ cation with sp² and sp anion[a]

| entry | cation | carbanion | product | total reaction time (s) | yield (%) |
|-------|--------|-----------|---------|-------------------------|-----------|
| 1 | 2a | CF₃ / Li | 14 | 5.2 | 89 |
| 2[b] | | F / Li | 15 | 6.3 | 92 |
| 3[b] | | NC / Li | 16 | 1.7 | 66 |
| 4[b] | | Br / Li | 17 | 6.3 | 76 |
| 5[b] | | MeO / Li | 18 | 6.3 | 77 |
| 6 | | MeO / Li | 19 | 5.2 | 83 |
| 7[b] | | OMe / Li | 20 | 6.3 | 72 |
| 8[c] | 2d | F / Li | 21 | 2.9 | 80 |
| 9 | 2e | F / Li | 22 | 6.3 | 85 |
| 10 | | NC / Li | 23 | 0.65 | 70 |
| 11 | 2b | F / Li | 24 | 6.3 | 73 |
| 12 | 2c | F / Li | 25 | 3.5 | 48 |
| 13 | 2a | nBu / Li | 26 | 5.3 | 96 |
| 14[b] | | MeO / Li | 27 | 3.9 | 80 |
| 15[b] | | epoxide-O / Li | 28 | 2.8 | 70 |
| 16[b] | | NC / Li | 29 | 3.9 | 90 |
| 17 | | thiophene / Li | 30 | 4.7 | 84 |
| 18 | | Me₃Si / Li | 31 | 4.5 | 89 |
| 19 | | Ph / Li | 32 | 3.5 | 92 |
| 20 | 2d | Me₃Si / Li | 33 | 3.5 | 92 |
| 21 | 2b | Ph / Li | 34 | 4.5 | 75 |
| 22 | 2f | Ph / Li | 35 | 5.4 | 61 |

[a]Flow microreactor composed of a V-shaped micromixer (inner diameter: 250 μm) and two T-shaped ones (inner diameter: 250 μm) was used. Yields were determined by GC. Typical reaction condition for aryl lithium: cation precursor (0.05 M), TfOH (0.05 M, 2.0 eq), anion precursor (0.18 M, 5.4 eq), and *n*BuLi (0.60 M, 4.5 eq) were reacted at −15 °C. Typical reaction condition for alkynyl lithium: cation precursor (0.05 M), TfOH (0.05 M, 2.0 eq), anion precursor (0.18 M, 4.3 eq), and *n*BuLi (0.60 M, 3.6 eq) were reacted at −15 °C. Typical reaction condition for thionium ion with alkynyl lithium: cation precursor (0.05 M), TfOH (0.20 M, 12 eq), anion precursor (0.50 M, 16 eq), and *n*BuLi (0.99 M, 16 eq) were reacted at 0 °C. See SI for detailed reaction conditions.
[b]The coupling reaction was carried out at 15 °C.

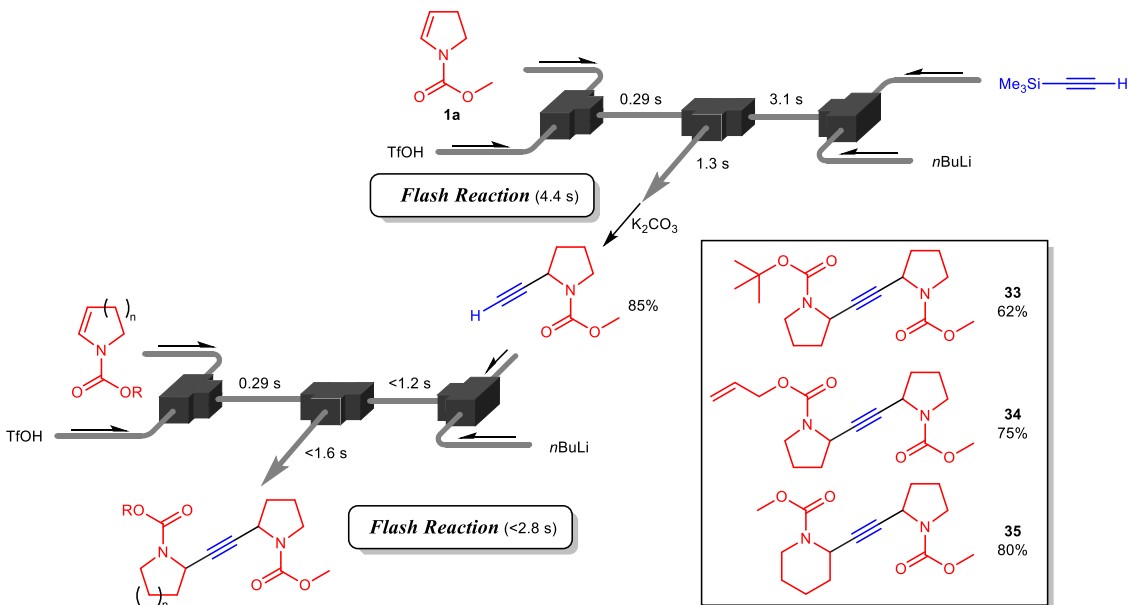

**Fig. 3 | Twice direct cross-coupling reaction using four intermediates.** [a] Flow microreactor system composed of two V-shaped micromixers (inner diameter: 250 μm) and four T-shaped ones (inner diameter: 250 μm) was used. Typical reaction condition for the first reaction: cation precursor (0.05 M), TfOH (0.05 M, 2.0 eq), trimethylsilyl acetylene (0.18 M, 4.3 eq), and $n$BuLi (0.60 M, 3.6 eq) at −10 °C. Typical reaction condition for the second reaction: cation precursor (0.05 M), TfOH (0.05 M, 2.0 eq), terminal alkyne (0.18 M, 8.6 eq), and $n$BuLi (0.60 M, 7.2 eq) at −10 °C.

reactions enabled the rapid synthesis of unsymmetrical alkynes bearing $N$-containing cyclic motifs. Four types of reactive intermediates were separately generated in different flow paths, and once mixed, their coupling reaction proceeded instantly. This system controls the flow reaction space, regulating the generation time and reactions of the reactive intermediates. Therefore, convergently integrated rapid reactions demonstrated a swift construction of molecular complexity, including alkynes bearing different pyrrolidines at both carbons.

## Discussion

In summary, the bio-inspired convergent approach of double intermediates was demonstrated by virtue of flow microreactors. Flow microreactor's capability of rapid and highly selective reactions enabled the synthetic method, in which acidic activation of unsaturated precursors irreversibly generates cationic species, preventing undesired dimerization reactions. Cationic species and carbanions were respectively generated in different parts of the flow system, and met together before they decompose, enjoying their high reactivities. It is anticipated that this series of "flash" reactions could achieve complicated syntheses by taking advantage of multiple reactive intermediates, and further investigation is currently underway.

## Methods

Flow microreactors are composed of stainless steel T- and V-shaped micromixers with inner diameter of 250 μm, stainless steel microtube reactors with 1000, 500, and 250 μm inner diameter, and PTFE tube with inner diameter of 1000 μm.

### Reactions of cationic species with neutral nucleophiles

A flow microreactor system consisting of a V-shaped micromixer (**M1**) and a T-shaped micromixer (**M2**), two microtube reactors (**R1** and **R2**), and three pre-cooling units (**P1**–**P3**) was used. The flow microreactor system was dipped in a cooling bath. A solution of cation precursor (0.0500 M in CH$_2$Cl$_2$) and a solution of TfOH (0.050 M in CH$_2$Cl$_2$) was introduced into **M1** using syringe pumps. The mixed solution was passed through **R1** and was mixed with a solution of the nucleophile (0.20 M in CH$_2$Cl$_2$) in **M2**. The resulting solution was passed through

**R2**. After a steady state was reached, an aliquot of the product solution was collected and treated with TBAF, Et$_3$N, and brine.

### Reactions of cationic species with carbanions

A flow microreactor system consisting of a V-shaped micromixer (**M1**), two T-shaped micromixers (**M2** and **M3**), three microtube reactors (**R1** and **R2**), and four pre-cooling units (**P1**–**P4**) was used. The flow microreactor system was dipped in a cooling bath. A solution of the cation precursor (0.0500 M in CH$_2$Cl$_2$) and a solution of TfOH (0.050 M in CH$_2$Cl$_2$) was introduced into **M1** using syringe pumps. The mixed solution was passed through **R1** to **M3**. Whereas, a solution of the anion precursor (0.18 M in THF) and a solution of $n$-BuLi (0.60 M in $n$-hexane) were introduced into **M2** using syringe pumps, and the mixed solution was passed through **R2** to **M3**. Those solutions are mixed in **M3**, and the resulting solution is passed through **R3**. After a steady state was reached, an aliquot of the product solution was collected and treated with brine.

### Reporting summary

Further information on research design is available in the Nature Portfolio Reporting Summary linked to this article.

## Data availability

Experimental procedures, characterization of new compounds, and all other data supporting the findings are available in the Supplementary Information. All data were available from the corresponding author upon request.

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

## Acknowledgements

This work was supported by JSPS KAKENHI Grant Numbers, JP20KK0121 (Y.A. and A.N., Fostering Joint International Research (B)), JP21H01936 (A.N., Grant-in-Aid for Scientific Research (B)), JP21H01706 (A.N., Grant-in-Aid for Scientific Research (B)), JP19K22186 (A.N., Grant-in-Aid for Challenging Exploratory Research), JP20K15276 (Y.A., Grant-in-Aid for Early-Career Scientists), and JP21H05080 (A.N., Grant-in-Aid for Transformative Research Areas (B)). This work was also partially supported by AMED (A.N., JP21ak0101156), the Core Research for Evolutional Science and Technology (CREST, A.N., JPMJCR18R1), New Energy and Industrial Technology Development Organization (NEDO, A.N., JPNP14004 and JNPN19004), the Japan Keirin Autorace Foundation (Y.A. and A.N.), and the Ogasawara Foundation for the Promotion of Science & Engineering (A.N.). The authors are grateful to Ms. Eriko Kusaka (Kyoto University) for her kind assistance of NMR analyses. The authors are indebted to Prof. Atsushi Minami (Tokyo Institute of Technology) for many discussions concerning biosynthesis. The authors express their sincere thanks to Dr. Shusaku Asano (Kyushu University) for helpful discussions

and comments about the quantification of the performance of flow microreactors.

## Author contributions

H.S. and A.N. conceived and designed the experiments. H.S., H.Y., Y.S., and M.T. conducted experiments. Y.A. and A.N. supervised the experiments. H.S., Y.A., and A.N. wrote the manuscript.

## Competing interests

The authors declare no competing interests.
