## [Peer Review File · Nature Communications]

Convergent approach for direct cross-coupling enabled by flash irreversible generation of cationic and anionic speciesReviewers' Comments:

Reviewer #1:

Remarks to the Author:

The manuscript entitled "Convergent approach of double intermediates in flow microreactor: flash irreversible carbocations generation enables direct cross-coupling" by Aiichiro Nagaki et al. report a convergent synthetic approach by independently generating carbocation and carbonation at once and allowing them to react with each other using flash chemistry resulting in C-C bond formation. The reaction took place within a few seconds which is not applicable in conventional reactions. However, the concept of unstable irreversible cation generation and its trapping by the nucleophile is not new and has recently been explored by the same group using a flash electrochemical approach to the carbocation. (Angew. Chem. Int. Ed. 2022, 61, e2021161). There are less significant improvement in science aspect, the additional scope of similar chemistry category which is not novel flash chemistry work. Besides, there are other methods available for the synthesis of target products which is possible in the batch reactor. Their comparison with the presented approach is missing in the manuscript. And it seems this work is not suitable to meet high standard of ACIE. Here are additional comments.

1) The manuscript repeatedly used the word carbocations, which creates the impression that the present methodology is applicable to all kinds of carbo-cations. However, it is limited to only oxy-carbenium and iminium ions. The author also focused only on oxy carbenium and iminium ions. Also, the observed less stability of oxy carbenium and iminium ions is due to the method of their generation.

2) The present manuscript does not cover all kinds of oxy carbenium and iminium ions. Such as asymmetric enamines for the iminium ion synthesis which are widely used in organic synthesis.

3) Table 1. Entry. 10, 11, and 12. It is surprising that organolithium nucleophile gave the highest yield in comparison to other organometallic nucleophile. The observed results do not match the principle of soft-to-soft and hard-to-hard cation and anion combination. Need explanation.

4) Present methods do not look like it is bio-inspired as they claim in the conclusion and in the introduction part. To successfully claim, need to add one image accordingly.

Reviewer #2:

Remarks to the Author:

In this manuscript, the authors describe their experimental work in synthetic organic chemistry. Specifically, they use flow chemistry in a micro-reactor system to carryout conversions with primarily N-acyliminium ions, but also with a carboxonium ion system. The methodology is capable of directly coupling cationic reactive intermediates and fairly strong nucleophiles, including carbanionic species. The authors examined the influence of flow rates and temperature to optimize the conversions.

There are some nice results in this manuscript, however it cannot be published in its present form. Here are some of the problems...

The authors suggest that this is a "biomimetic" synthesis. How so? They use generalities, such as pointing out enzymes utilize reactive intermediates in biosynthetic reactions. In the opinion of this reviewer, that does not qualify the methodology as biomimetic. Their suggestion might have some weight if they could actually give one or two specific examples of biosynthetic chemistry that is similar to the reported work.

The authors incorrectly use the term carbocation (even in the title). In general, iminium ions are not considered "carbocations," even though a minor resonance structure can be drawn showing a positive charge on a carbocation. The paper should reframe the discussions to describe coupling of the N-acyl iminium ions with strong nucleophiles. This is still a nice reaction – the authors do not need to try and sell the work as carbocation chemistry.

The pKa of triflic acid in MeCN is not likely 2.6. The authors do not provide a reference for this number, so it is not clear where the number comes from.

In the batch experiments, the authors use an excess of TfOH to generate the iminium ion (0.5 equivalents excess). This probably causes a decrease in the yield of the batch reactions, as the allyl silane is not stable to the superacid. Since a premise of the paper is that flow works better than batch, this may be an unfair comparison. The authors need to do a reasonably good effort to run a good batch reaction, if they want to make this comparison.

Some other issues or corrections:

Ref 2 – no title

Page 1 – The two statements contradict: paragraph 2, sentence 1, “short-lived anionic species...” and paragraph 3, sentence 3, “...less stable than carbanions.” In one comment, the authors are saying that carbanions are unstable, and then they say, they are not too unstable.

Page 2 – The authors refer to cationic polymerization. Of what? Do they observe solids? A little more commentary would be helpful.

Page 3 – The contour maps should have Product Yield along the top of the graphic. Readers are forced to search the legend to see what the numbers represent.

Page 4 – lower yields from higher temperatures and longer reaction time may also be due to secondary reactions of the products and excess acid – not necessarily due to the “decomposition of the carbocation.”

Table 1, entries 9-12 – these are likely highly exothermic reactions; the authors should comment on temperature control (an advantage for flow systems).

Page 6 – the authors use the terms “highly ionic counteranions” and “highly ionic cations.” These are odd terms. In the case of cations, this concept is generally associated with low coordinating counterions, not “highly ionic.”

Page 6 – the authors should mention how the aryl anions are being generated, rather than forcing readers to examine the Supporting Information.

Figure 3 and accompanying text – from the Supporting Information, it appears that intermediate coupling product is isolated, deprotected with MeOH and K₂CO₃ in a batch reactor, and then subjected to the second coupling step. The authors do not mention this in the text and suggest in the figure that it goes straight through in the flow system. This part of the procedure needs to be specifically stated, so as to not mislead readers.

Authors do not provide an analysis of the relative concentrations or equivalents of reagents mixing at each stage. Given f₁, f₂, f₃, and the concentrations of the reagents, what are the ratios of the reagents when they are mixed.

Reviewer #3:

Remarks to the Author:

In this work, the authors use the rapid transport rates of flow chemistry to quickly transport reactive carbocation and carbanion intermediates to perform direct C-C bond formation before the intermediates could decompose. The carbocations were generated in flow by the addition of trifluoromethylsulfonic acid to an enamine and were detected by reaction with allyltrimethyl silane. In the flow reactor, it was determined that high flowrates improved the reaction conversion due to improved mixing and transport with low flowrates leading to undesired polymerization side products. The carbocation reaction was additionally utilized for a range of nucleophiles (allyl, oxy silanes, alkyl Zn Mg and Li). Direct in flow C-C bond formation was performed by running both the carbocation and carbanion synthesis steps immediately prior to mixing and creating the coupled product. Alkyl and aryl lithium intermediates were selected as the carbanion species affording good coupling yields 48-

96% with a very rapid reaction time <7 s. The rapid intermediate generation was also successfully extended to a telescoped reaction with the coupled product being the carbanion precursor of the second stage after neutralization of the residual acid.

Overall, the presented work is interesting but lacks details regarding the mass transport characteristics of the flow reactor. Mass transfer characterization is important to better understand the performance of the flow reactor. Therefore, the manuscript in its current form cannot be recommended for publication in Nature Communications.

Major Comments:

1. Have the authors attempted to decouple mixing from residence time? (e.g. using the same flowrate through the mixing junction and changing mixing tube length or fixed residence time with different flowrates for each tube length). In the current configuration, the mixing and residence times are interrelated.
2. Was an estimate of the flow regime in the micromixer performed? (Reynold's Number / laminar vs. turbulent flow) It is very important to report such dimensionless numbers.
3. Is there a sense of the relative rate of mass transport vs. reaction kinetics? (i.e., Damkohler Number)

Print Email

REVIEWER COMMENTS

Reviewer #1:

The manuscript entitled "Convergent approach of double intermediates in flow microreactor: flash irreversible carbocations generation enables direct cross-coupling" by Aiichiro Nagaki et al. report a convergent synthetic approach by independently generating carbocation and carbonation at once and allowing them to react with each other using flash chemistry resulting in C–C bond formation. The reaction took place within a few seconds which is not applicable in conventional reactions.

We appreciate your reviewing.

However, the concept of unstable irreversible cation generation and its trapping by the nucleophile is not new and has recently been explored by the same group using a flash electrochemical approach to the carbocation. (*Angew. Chem. Int. Ed.* 2022, 61, e2021161). There are less significant improvement in science aspect, the additional scope of similar chemistry category which is not novel flash chemistry work.

We agree that the concept of irreversible generation of unstable cation and its trapping by the nucleophile has a commonality with our previous ACIE report. However, the present chemical approach is totally different from the electrochemical generation, for example, no need for a special electrochemical device. The time-range is much shorter. To clarify it, we added the following sentences to the manuscript.

“Among a variety of methods for generating carbocationic species, the use of strong acid may provide a rapid generation of those unstable species without the specific equipment such as flow electrolysis devices.”

Besides, there are other methods available for the synthesis of target products which is possible in the batch reactor. Their comparison with the presented approach is missing in the manuscript.

Most products in our manuscript are novel compounds, and in particular, this is the first report synthesizing unsymmetrical alkynes connecting to pyrrolidines, the products shown in Figure 3. To clarify this, we added the following sentences in the manuscript.

“Therefore, convergently integrated rapid reactions demonstrated a swift construction of molecular complexity, including alkynes bearing different pyrrolidines at both carbons. To the best of our knowledge, this is the first example of synthesizing such unsymmetrical alkynes.”

1) The manuscript repeatedly used the word carbocations, which creates the impression that the present methodology is applicable to all kinds of carbocations. However, it is limited to only oxy-carbenium and iminium ions. The author also focused only on oxy carbenium and iminium ions.

We agree that repeated use of the word “carbocation” may lead mistake, although the concept itself is for the reaction of carbocations including oxonium and iminium ions with carbanions. To avoid the misleading the readers, we revised some of the word “carbocation” to other words such as “iminium ion” and “carbocationic species”.

Also, the observed less stability of oxy carbenium and iminium ions is due to the method of their generation.

We agree that the stability of the oxonium and iminium ions depends on the method of their generation. However, acidically-generated cations from unsaturated starting materials only can be utilized for reactions under the harsh condition; in the presence of strong acid. To clarify it, we investigated the reactions using weaker acids, and added the following sentences to the manuscript.

“It is noteworthy that the irreversible generation of the iminium ion is crucial for its reaction. When using weaker acids such as benzoic acid (pKa 22), acetic acid (pKa 24), and trifluoroacetic acid (pKa 12), their reactions with **1a** and allyltrimethylsilane in the flow microreactor resulted in recovery of the starting material (**1a**), and **3** was not obtained (see supporting information).”

2) The present manuscript does not cover all kinds of oxy carbenium and iminium ions. Such as asymmetric enamines for the iminium ion synthesis which are widely used in organic synthesis.

We appreciate your suggestion, and actually we now are interested in asymmetric onium ions. As we have already reported that asymmetric oxonium ions generated from TfOH with glycals can react with anionic species (reference 10), the asymmetric iminium ions from asymmetric amines such as proline would be applicable for this transformation. Instead of oxocarbenium and iminium cations, we investigated the generation of a thionium ion, and found it can be generated in this system to react with anionic species. Thus, we added this entry (Table 2, entry 22) as well as the following sentence in the manuscript.

“As well as the iminium ions, this integrated flow system allowed generation of thionium ion **2f**, which reacted with sp anion to afford the coupling product in a good yield (entry 22).”

3) Table 1. Entry. 10, 11, and 12. It is surprising that organolithium nucleophile gave the highest yield in comparison to other organometallic nucleophile. The observed results do not match the principle of soft-to-soft and hard-to-hard cation and anion combination. Need explanation.

Since the cationic species decomposed within seconds (shown in Figure 2), reactive nucleophile, organolithium reagents are preferred because they can rapidly react with electrophiles. Whereas, the reactions of other nucleophiles with the cationic species competes with their decomposition. To clarify it, we revised the manuscript as follows.

“As the electronegativity of the counteranion increased (Li:0.98, Mg:1.31, and Zn:1.65), the reaction yield decreased, suggesting that reactive anions are suitable for direct reactions with the cationic species... This suggests that because of its high reactivity, n-butyl lithium could react with the onium cation before decomposition.”

4) Present methods do not look like it is bio-inspired as they claim in the conclusion and in the introduction part. To successfully claim, need to add one image accordingly.

Although our concept using multiple reactive species with short lifetime is inspired by bio-synthesis referred as ref 2 and 3, we agree that our research is not biomimetic. To avoid misleading the readers, we revised the manuscript removing the word “biomimetic”.

Reviewer #2:

In this manuscript, the authors describe their experimental work in synthetic organic chemistry. Specifically, they use flow chemistry in a micro-reactor system to carry out conversions with primarily N-acyliminium ions, but also with a carboxonium ion system. The methodology is capable of directly coupling cationic reactive intermediates and fairly strong nucleophiles, including carbanionic species. The authors examined the influence of flow rates and temperature to optimize the conversions.

We appreciate your reviewing.

There are some nice results in this manuscript, however it cannot be published in its present form. Here are some of the problems...The authors suggest that this is a “biomimetic” synthesis. How so?...

We agree that our results are not “biomimetic”. Thus, we revised the manuscript removing the word “biomimetic”.

The authors incorrectly use the term carbocation (even in the title). In general, iminium ions are not considered “carbocations,” even though a minor resonance structure can be drawn showing a positive charge on a carbocation. The paper should reframe the discussions to describe coupling of the N-acyl iminium ions with strong nucleophiles. This is still a nice reaction – the authors do not need to try and sell the work as carbocation chemistry.

We appreciate the reviewer mentioning “This is still a nice reaction”. Although IUPAC defines onium ions as carbocations, we agree some chemists do not consider so. According to the reviewer’s comment, and to avoid misleading readers, we revised the manuscript to use the words “carbocationic species”, “iminium and oxonium ions” instead of “carbocations” in most parts.

The pKa of triflic acid in MeCN is not likely 2.6. The authors do not provide a reference for this number, so it is not clear where the number comes from.

We added the following reference as reference [11]: Leito, I. *et al.* pKa values in organic chemistry – Making maximum use of the available data. *Tetrahedron Lett.* **2018**, *59*, 3738.

In the batch experiments, the authors use an excess of TfOH to generate the iminium ion (0.5 equivalents excess). This probably causes a decrease in the yield of the batch reactions, as the allyl silane is not stable to the superacid. Since a premise of the paper is that flow works better than batch, this may be an unfair comparison. The authors need to do a reasonably good effort to run a good batch reaction, if they want to make this comparison.

We appreciate this suggestion, and investigated the batch reactions using 1.0 eq of TfOH, however, the yields were dramatically decreased. We added the following sentence to the manuscript.

“The batch reactions with 1.0 equivalent of TfOH also resulted in low yields (see supporting information).”

Some other issues or corrections:

Ref 2 – no title

We revised the manuscript.

Page 1 – The two statements contradict: paragraph 2, sentence 1, “short-lived

anionic species..." and paragraph 3, sentence 3, "...less stable than carbanions." In one comment, the authors are saying that carbanions are unstable, and then they say, they are not too unstable.

We deleted the latter statement to avoid misleading the readers.

Page 2 – The authors refer to cationic polymerization. Of what? Do they observe solids? A little more commentary would be helpful.

Actually, we thoroughly analyzed the reaction mixture again, and found dimer **4**. This proves that with a bad-mixing condition, the unreacted enamines work as nucleophiles, reacting the generated iminium ion. The amount of the dimer correlates strongly with the mixing efficiency. Thus, we revised the manuscript to clarify this tendency by adding the following sentences, as well as Figure 2c, and removed sentences referring to cationic polymerization. We totally appreciate this comment because these findings proved the importance of flash generation of the cationic species.

"The best results under the batch condition (Method C, $-40\text{ }^{\circ}\text{C}$) showed non-negligible amount of **4**, which is derived from the reaction of **1a** with **2a**, whereas the flow condition (V250, 20 mL/min) showed less amount of **4** (Fig. 2c). Thus, these investigations proved that the reactions with short mixing time can trigger the desired reaction between TfOH and enamine **1a** before it reacts with **2a**."

Page 3 – The contour maps should have Product Yield along the top of the graphic. Readers are forced to search the legend to see what the numbers represent.

We added the title of the contour map.

Page 4 – lower yields from higher temperatures and longer reaction time may also be due to secondary reactions of the products and excess acid – not necessarily due to the "decomposition of the carbocation."

We added the following underlined sentence to the manuscript.

"Additionally, the lower yield at higher temperatures and longer reaction times could be derived from the decomposition of the iminium ion, such as the β -elimination of the generated iminium to afford the starting enamine, as well as the secondary reactions of the products with the excess acid."

Table 1, entries 9-12 – these are likely highly exothermic reactions; the authors should comment on temperature control (an advantage for flow systems).

We really appreciate your comment, and added the following underlined sentence in the manuscript

“Direct cation-anion coupling afforded the desired product **12**, where an sp^3-sp^3 C–C bond was formed in high yield by virtue of the fast mixing and precise time control together with the precise temperature control of the flow systems.”

Page 6 – the authors use the terms “highly ionic countercations” and “highly ionic cations.” These are odd terms. In the case of cations, this concept is generally associated with low coordinating counterions, not “highly ionic.”

We removed the terms “highly ionic” in the manuscript.

Page 6 – the authors should mention how the aryl anions are being generated, rather than forcing readers to examine the Supporting Information.

The following sentence was added in the manuscript to briefly explain how the aryl anions were generated

“As carbanions, aryl lithiums, which were generated from the corresponding aryl bromides with nBuLi in the flow microreactor, were utilized for this transformation.”

Figure 3 and accompanying text – from the Supporting Information, it appears that intermediate coupling product is isolated, deprotected with MeOH and K_2CO_3 in a batch reactor, and then subjected to the second coupling step. The authors do not mention this in the text and suggest in the figure that it goes straight through in the flow system. This part of the procedure needs to be specifically stated, so as to not mislead readers.

We apologize that Figure 3 and text led to misleading readers. In Figure 3, we clarified that the intermediate was isolated. Meanwhile we revised the manuscript so as not to mislead readers.

“The first alkynylation reaction proceeded rapidly, and after deprotection and isolation, the next alkynylation completed within 2.8 s after the activation of the precursor. This series of reactions enabled the rapid synthesis of unsymmetrical alkynes bearing N-containing cyclic motifs.”

Authors do not provide an analysis of the relative concentrations or equivalents of reagents mixing at each stage. Given f_1 , f_2 , f_3 , and the concentrations of the reagents, what are the ratios of the reagents when they are mixed.

We added the information of reaction conditions, the concentration and equivalent of the reagents, at the footnote of Table 2 and Figure 3.

Reviewer #3:

In this work, the authors use the rapid transport rates of flow chemistry to quickly transport reactive carbocation and carbanion intermediates to perform direct C-C bond formation before the intermediates could decompose... The rapid intermediate generation was also successfully extended to a telescoped reaction with the coupled product being the carbanion precursor of the second stage after neutralization of the residual acid.

We appreciate your reviewing.

Overall, the presented work is interesting but lacks details regarding the mass transport characteristics of the flow reactor. Mass transfer characterization is important to better understand the performance of the flow reactor.

Although our group is interested in organic synthesis, we regret the lack of details regarding the mass transport characteristics. Based on our previous results, such as the reaction control within 3 milliseconds, we have believed that the mass transport of our reaction system completes instantaneously. We pause to realize the necessity for the investigation about the mass transport, focusing on the mixing efficiency of the micromixer because it is a dominant factor of the mass transport of flow reactors. We made every effort to gain the information of the mixing efficiency, such as Villiermaux–Dushman method to estimate the mixing time. We revealed the reaction condition with lower yield has slow mixing time (longer than 1 ms), whereas that leading to high yield has fast mixing time (less than 0.1 ms). We added the following sentences to the manuscript. Also, we added a new section in supporting information describing mass transfer characterization.

“For a deeper understanding of the mixing efficiency, we explored the Villiermaux–Dushman method, which assesses mixing efficiency of micromixers and estimates mixing time of aqueous medium. Based on Yin’s condition, we estimated the mixing time of the solutions introduced into the V-shaped 250 μm micromixer (Fig. 2b, dots). The results clearly showed the difference; the condition with the smallest flow rate (2.5 mL/min) had longer mixing time (more than 1 millisecond), whereas that of other conditions was less than 0.1 millisecond.”

The following answers are based on Villiermaux–Dushman method. We appreciate the comments, which improved our research not only in the field of synthetic chemistry but also that of chemical engineering. We believe the followings must answer to your comments.

1. Have the authors attempted to decouple mixing from residence time? (e.g. using the same flowrate through the mixing junction and changing mixing tube

length or fixed residence time with different flowrates for each tube length). In the current configuration, the mixing and residence times are interrelated.

We agree the necessity for decoupling mixing from residence time. Thus, we retried the initial investigation, changing the volume of tube reactor to maintain the residence time (0.042 s), and revised Figure 2b. Additionally, we investigated the same reactions using mixers having lower mixing efficiency. The results shown in Figure 2b indicates high flow rate, which leads to a good-mixing condition, and mixers having higher mixing efficiency is beneficial for this reaction, clarifying the importance of mixing. We added the following sentence in the manuscript.

“This indicates that the condition improving the mixing efficiency is beneficial for this transformation, which was supported by the reactions using other micromixers. The reactions with a thinner and sharper-angled micromixer, which has a better mixing efficiency, showed higher yield.”

2. Was an estimate of the flow regime in the micromixer performed? (Reynold’s Number / laminar vs. turbulent flow) It is very important to report such dimensionless numbers.

We calculated Reynold’s Number of the best condition, and it was about 2200 indicating this condition is in a transition to a turbulent flow regime. We added the detail of this calculation in supporting information.

3. Is there a sense of the relative rate of mass transport vs. reaction kinetics? (i.e., Damkohler Number)

Although we need a detailed analysis such as CFD simulation, we estimated Damköhler number of the reaction as less than 0.1. Thus, we added the following sentences in the manuscript.

“Although an advanced analysis such as CFD simulation must be necessary to gain the accurate one, we calculated the Damköhler number of these conditions to be below 0.1 (see supporting information). This indicates that the mass transport is enough faster than the reaction, which is supported by the fact that higher flow rate than 10 mL/min did not change the product yields.”

Additionally, the detail of this calculation is described in supporting information, as follows.

“Although the value should have a margin of error, the scale of both the reaction time and the mixing time of the cation generation reaction can be estimated. Since Table S4 (*vide infra*) indicates the reaction at 0 °C reached the highest yield with 82 ms of the residence time, reaction time is in 10 ms scale. Whereas, above-mentioned Villermaux–Dushman protocol indicates the mixing time for high-yielding condition (total flow rate is higher than 2.5 mL/min) must be smaller than

1 ms. Thus, D_a can be determined as smaller than 0.1, which is much smaller than 1.”

Reviewers' Comments:

Reviewer #2:

Remarks to the Author:

In this manuscript, the authors describe their experimental work in synthetic organic chemistry. Specifically, they use flow chemistry in a micro-reactor system to carry out conversions with primarily N-acyliminium ions, but also with a carboxonium ion system. The methodology is capable of directly coupling cationic reactive intermediates and fairly strong nucleophiles, including carbanionic species. The authors examined the influence of flow rates and temperature to optimize the conversions.

The authors completed some important revisions, however there are still some problematic issues. Consequently, publication is not recommended.

As previously mentioned, the authors insist on using the term "carbocation" for their chemistry with N-carbamoyl iminium ions. Contrary to their response (to reviewer comments), this reviewer found no evidence from IUPAC publications suggesting that these species should be called carbocations. And since the authors did not provide their source of this nomenclature, I am highly skeptical of this claim. Perhaps the leading authority on nucleophile-electrophile reactions is Professor Herbert Mayr. When his publications are examined, these species are never called carbocations, but rather they are consistently described as iminium ions. Moreover, the authors cite Yoshida's closely related work with cation pool chemistry (reference 13) and Yoshida does not refer to these ions as "carbocations." I cannot recommend publication of this manuscript if the authors continue to use incorrect nomenclature.

Some other items:

1. Line 42-43 – the authors claim that there are limited methods for generating carbocations (or is it iminium ions?). This is not really true and the authors should have been a little more diligent in their scholarship before making this claim. A brief examination of a good physical organic chemistry book would reveal a wide variety of cation generating reactions, both Bronsted and Lewis acid promoted conversions, chemical methods (i.e., diazotization), and others (such as oxidation routes).
2. Line 43 – "small number" is better stated low concentration.
3. Line 79 – "the low yields are due to unsuccessful generation of 2a" This is a presumption. There are other reasons for low yields and the authors really do not support their statement.
4. Line 132 – The authors suggest deprotonation of the iminium ion as an explanation for low yields. Is unreacted starting material (enamine) found in the product mixture or are there increasing amounts of dimers at high temperatures/long reaction times?
5. Line 141 and 142 – Again, this conclusion is not well supported. The authors suggest that the contour plot – which shows lower yields for the piperidine system – indicates that the piperidine iminium ion is less stable than the pyrrolidine iminium ion. Then they suggest the flow systems reveal the nature of the reactive intermediates. However, there may be other reasons that the yields are low for one reaction versus another, reasons that have nothing to do with the stability of a proposed intermediate. In fact, the reverse could even be true. Perhaps the stability of the piperidine iminium ion slows the reaction with the allyl silane. Thus, it is a bit naïve to suggest that the nature of reactive intermediates can be obtained from the data from a flow system. It is likely much more complicated.
6. Figure 2 and lines 143-146 – the reported low temperature NMR spectrum of ion 2a needs some explanation. The authors need to explain why the ¹H NMR integrations appear to be off. For example, the integration of Ha should be 1.0. However, the methoxy proton signal (assumed at ca. 5.3 ppm) are not nearly at the expected integration of 3.0. It looks like an integration of about 1.3, maybe. A similar situation arises for some of the methylene protons.
7. Line 183 – should "via" actually be "without"
8. Line 187 – counter ion
9. Line 190 – there are several explanations for varying reactivities of organometallic, including aggregation states in particular solvents.
10. Lines 206-207 – this sentence needs to be restated. It does not make sense.

Reviewer #3:

Remarks to the Author:

The authors have properly addressed all questions and concerns raised by the reviewer.

REVIEWER COMMENTS

Reviewer #2:

As previously mentioned, the authors insist on using the term "carbocation" for their chemistry with N-carbamoyl iminium ions. Contrary to their response (to reviewer comments), this reviewer found no evidence from IUPAC publications suggesting that these species should be called carbocations. And since the authors did not provide their source of this nomenclature, I am highly skeptical of this claim.

As you rightfully pointed out, we have thoroughly reconsidered the use of the term 'carbocation' in reference to our chemistry involving N-carbamoyl iminium ions. After a careful reevaluation of your concerns and in alignment with your suggestion, we have revised our manuscript to eliminate the use of the term 'carbocation' entirely. We appreciate your diligence in bringing this matter to our attention.

In response to your skepticism regarding the nomenclature, we acknowledge the importance of clarity and accuracy in scientific terminology. Although our initial reference may not have been explicit, we have since revised our manuscript to reflect the appropriate and accurate terminology in line with IUPAC recommendations. Thank you for your valuable feedback, and we hope these revisions address your concerns.

Some other items:

1. Line 42-43 – the authors claim that there are limited methods for generating carbocations (or is it iminium ions?). This is not really true and the authors should have been a little more diligent in their scholarship before making this claim. A brief examination of a good physical organic chemistry book would reveal a wide variety of cation generating reactions, both Bronsted and Lewis acid promoted conversions, chemical methods (i.e., diazotization), and others (such as oxidation routes).

Thank you for your thoughtful comments. We have revised the manuscript to include mentions of various cation-generating methods, including diazotization and chemical oxidation, and to add appropriate references. We appreciate your valuable input in improving the manuscript.

2. Line 43 – "small number" is better stated low concentration.

We have considered your comments carefully and made revisions to the manuscript accordingly.

3. Line 79 – "the low yields are due to unsuccessful generation of 2a" This is a presumption. There are other reasons for low yields and the authors really do not support their statement.

Thank you for your detailed feedback. Upon reevaluation, we have identified that the low yields were not solely attributed to the unsuccessful generation of 2a but rather resulted from incomplete mixing, leading to dimerization (iminium ion 2a reacts with enamine 1a). The evidence supporting this includes the observed formation of dimer 4. Taking this into consideration, we have revised the manuscript accordingly to reflect this clarification.

4. Line 132 – The authors suggest deprotonation of the iminium ion as an explanation for low yields. Is unreacted starting material (enamine) found in the product mixture or are there increasing amounts of dimers at high temperatures/long reaction times?

Thank you for your insightful question. In our reexamination, although we did not detect the enamine in the product mixture, we observed an increased presence of dimer 4 at higher temperatures and longer reaction times. Since 4 results from the reaction between the enamine with the iminium cation, this suggests an enhanced generation of enamine under these conditions. We have incorporated this observation into our revised manuscript to provide a more accurate representation of the reaction outcomes.

5. Line 141 and 142 – Again, this conclusion is not well supported. The authors suggest that the contour plot – which shows lower yields for the piperidine system – indicates that the piperidine iminium ion is less stable than the pyrrolidine iminium ion. Then they suggest the flow systems reveal the nature of the reactive intermediates. However, there may be other reasons that the yields are low for one reaction versus another, reasons that have nothing to do with the stability of a proposed intermediate. In fact, the reverse could even be true. Perhaps the stability of the piperidine iminium ion slows the reaction with the allyl silane. Thus, it is a bit naïve to suggest that the nature of reactive intermediates can be obtained from the data from a flow system. It is likely much more complicated.

That is a valid point, and we appreciate your insight. In response, we have refrained from making conclusive statements about the stability of intermediates and have focused solely on presenting experimental facts; 3 was obtained in higher yield at higher temperatures and longer reaction times. However, we acknowledge the intriguing aspect of intermediate stability, and we intend to continue our exploration in this direction for future work. We are grateful for your valuable feedback, and your considerations will undoubtedly contribute to the refinement of our research.

6. Figure 2 and lines 143-146 – the reported low temperature NMR spectrum of ion 2a needs some explanation. The authors need to explain why the ¹H NMR integrations appear to be off. For example, the integration of Ha should be 1.0. However, the methoxy proton signal (assumed at ca. 5.3 ppm) are not nearly at the expected integration of 3.0. It looks like an integration of about 1.3, maybe. A similar situation arises for some for the methylene protons.

We appreciate for bringing attention to the concern regarding the low-temperature NMR spectrum of ion 2a. To enhance clarity, we omitted some details in the figure, which might have led to the confusion in NMR integrations. In response to your suggestion, we have revised the figure to include the integrations for better transparency. Additionally, we must acknowledge an oversight in the assignment of the solvent (dichloromethane) during the preparation of the figure. This misassignment may have contributed to the misunderstanding. We have rectified this error to ensure accurate representation in the revised version. We appreciate your diligence in reviewing our work, and these modifications should address the issues you raised.

7. Line 183 – should “via” actually be “without”

8. Line 187 – counter ion

We have considered your comments carefully and made revisions to the manuscript accordingly.

9. Line 190 – there are several explanations for varying reactivities of organometallic, including aggregation states in particular solvents.

According to this consideration, we have refrained from making speculative statements and have opted to present only the experimental facts.

10. Lines 206-207 – this sentence needs to be restated. It does not make sense.

We have considered your comments carefully and made revisions to the manuscript accordingly.

Reviewer #3:

The authors have properly addressed all questions and concerns raised by the reviewer.

We appreciate your reviewing and comments.

Reviewer #1:

However, the concept of unstable irreversible cation generation and its trapping by the nucleophile is not new and has recently been explored by the same group using a flash electrochemical approach to the carbocation. (Angew. Chem. Int. Ed. 2022, 61, e2021161). There are less significant improvement in science aspect, the additional scope of similar chemistry category which is not novel flash chemistry work.

In addition to our previous revisions, we would like to emphasize further the distinctions between our current work and the electrochemical approach in our publication (ACIE 61, e2021161 (2022)). The crucial difference lies in the fact that the electrochemically generated cation did not react with carbanion (*n*BuLi). Therefore, our approach, involving the generation of cations, stands as a novel concept, distinctly separate from the previous electrochemical study. We have included the following sentence as footnote 37 in the manuscript to highlight this distinction.

“We tested the reaction of anodically generated 1a with *n*-butyllithium in a batch reactor, and found no desired product 13, presumably because that the abundance of the supporting electrolyte in the surroundings led to the side reaction with *n*BuLi. See the supporting information for detail.”

Reviewers' Comments:

Reviewer #4:

Remarks to the Author:

[Note from the Editor: Reviewer #4 was asked to look over the response given to reviewer #2]

The authors described the "flash" generation of cation species (previously mentioned as carbocations) from enamine derivatives and TfOH. The short mixing time should be important to control the generation of the cation species, and it was accomplished by the flow technique. The generated cation species could be captured by nucleophiles, including aryl lithium, which would be relatively unstable. The authors could control the higher reactivity through a flow system, and the methodology enables high-selective reactions without any complicated activation of the starting materials. This reviewer suggests that the manuscript is acceptable in this journal after the minor revision summarized as below:

1. In the introduction section, the authors mentioned about the "bio-inspired" synthesis and described "organic chemists strive to emulate these conditions as closely as possible". However, this reviewer could not understand the similarity between the biosynthesis and the reaction described in the article. Of course, the biosynthesis can control the higher reactivity, but the strategy should be totally different. If the author would like to mention about these point, rational explanation should be stated.
2. The authors performed the cation generation using a batch system, but the reaction did not proceed efficiently. In Page 2, line 75, the following sentence was placed: "The reaction temperature and methods for adding reactants did not significantly affect the reaction yield." However, this reviewer supposes that the mixing efficiency should be important but the other terms, such as the instability, would not affect the reactivity. Thus, if enamine 1a was added to the solution of TfOH, the acidic activation should occur rapidly, and the dimerization could be prevented. How did the authors try using a syringe pump for the addition of enamine 1a or adding a diluted solution of 1a?
3. In the SI, the yield of the dimerization product using the batch system was not described. It should be helpful for the readers.

Reviewer #4:

The authors described the "flash" generation of cation species (previously mentioned as carbocations) from enamine derivatives and TfOH. The short mixing time should be important to control the generation of the cation species, and it was accomplished by the flow technique. The generated cation species could be captured by nucleophiles, including aryl lithium, which would be relatively unstable. The authors could control the higher reactivity through a flow system, and the methodology enables high-selective reactions without any complicated activation of the starting materials. This reviewer suggests that the manuscript is acceptable in this journal after the minor revision summarized as below:

We appreciate your reviewing and comments.

1. In the introduction section, the authors mentioned about the "bio-inspired" synthesis and described "organic chemists strive to emulate these conditions as closely as possible". However, this reviewer could not understand the similarity between the biosynthesis and the reaction described in the article. Of course, the biosynthesis can control the higher reactivity, but the strategy should be totally different. If the author would like to mention about these point, rational explanation should be stated.

Thank you for your detailed feedback. Indeed, our concept of using multiple reactive species generated in different space was inspired by the biosynthesis mentioned in references 2 and 3, but we agree that our work does not appear to be an exact imitation of the biosyntheses. Thus, to avoid misleading the readers, we have changed the text in the relevant sections of the manuscript.

2. The authors performed the cation generation using a batch system, but the reaction did not proceed efficiently. In Page 2, line 75, the following sentence was placed: "The reaction temperature and methods for adding reactants did not significantly affect the reaction yield." However, this reviewer supposes that the mixing efficiency should be important but the other terms, such as the instability, would not affect the reactivity. Thus, if enamine **1a** was added to the solution of TfOH, the acidic activation should occur rapidly, and the dimerization could be prevented. How did the authors try using a syringe pump for the addition of enamine **1a** or adding a diluted solution of **1a**?

We appreciate for bringing attention to the concern regarding the addition procedure of enamine **1a**. As you pointed out, the mixing efficiency must affect the reaction. Actually, when **1a** (or **1a** with allyltrimethylsilane) was added to TfOH (Method A and C), the desired product was obtained in moderate yields. Thus, we added the following sentences to the manuscript.

"The reactions of **1a** with TfOH in the presence of allyltrimethylsilane showed the similar results (Fig. 2, Method C and D). However, altering the sequence of reagent addition affected the result: method A and C, where the solution of **1a** was added to the solution of TfOH, resulted in the higher yield of **3** than the reverse-drop method (Method B and D) respectively. This tendency suggests the mixing efficiency affects the reaction efficiency."

Additionally, we revised the supplementary information to clarify how the solutions were added (the solutions were added for 30 seconds by hand).

3. In the SI, the yield of the dimerization product using the batch system was not described. It should be helpful for the readers.

Thank you for your kind comments. As you indicated, we showed the chapter named as “isolation and quantification of the dimer” in SI section 2.1 including the yields of the dimer in Table S2. Additionally, we added the following underlined description to the manuscript.

“Actually, the reaction of enamine 1a with TfOH in a batch reactor followed by the addition of allyltrimethylsilane afforded a small amount of the desired product **3**, and dimer **4** was obtained as the major product at each reaction temperature (Fig. 2, Method A and B, for yields of dimer 4 see Supplementary Table S2 for detail).”